# Evaluating the implementation of an early supported discharge (ESD) program for stroke survivors: A mixed methods longitudinal case study

**Danielle Hitch**[1,2]*, **Kathleen Leech**[1], **Sharon Neale**[1], **Avetta Malcolm**[1]

**1** Allied Health, Western Health, Sunshine, Victoria, Australia, **2** Occupational Therapy, Deakin University, Geelong, Victoria, Australia

* Danielle.Hitch@wh.org.au

**Data Availability Statement:** Data cannot be shared publicly because of the potential risk of identification inherent within case study research. Data are available from the Western Health Ethics

## Abstract

### Background

Early supported discharge (ESD) models of care for stroke survivors coordinate inpatient and community services, with the aim of reducing length of stay. While there is an established evidence base around the clinical outcomes of ESD), less is known about the implementation of this approach into existing stroke care service. The aim of this case study was to describe staff perceptions of the implementation of an ESD model of care for stroke survivors at a large metropolitan public hospital in Australia.

### Methods

This case study utilised a mixed methods design, which was designed in explicit alignment with the Consolidated Framework for Implementation Research (CFIR). Participants included staff that referred patients for ESD, and staff involved in the planning, implementation or delivery of ESD. Survey data was collected at three time points (ESD commencement, 3 months and 6 months), and focus groups were undertaken at the conclusion of the study. All quantitative data was analysed descriptive, while qualitative data was evaluated using thematic analysis.

### Results

Results from both sources of data identified changes in staff perceptions of ESD implementation over time. While very few changes were statistically significant, they were diverse patterns of change across the CFIR constructs over time. The characteristics of individuals and ESD characteristics attracted consistently positive perceptions, while patient needs and resources was the most prevalent theme within the data. While perceptions of factors related to the inner setting were mixed, there was a steady improvement in perceptions about the process across the later stages of implementation.

Committee (contact via Western Health) for researchers who meet the criteria for access to confidential data. Data can be request from the Office of Research, Western Health, Level 3 Western Centre for Health, Research & Education (CHRE), Sunshine Hospital, Furlong Road, St Albans Victoria 3021, Australia.

**Funding:** The author(s) received no specific funding for this work.

**Competing interests:** The authors have declared that no competing interests exist.

## Conclusions

The sophistication of knowledge translation and implementation in modern complex health-care environments is highlighted by the multiple interactions between the CFIR domains and constructs. While the implementation process described was generally positive and effective, using the CFIR as a framework confirmed that it also entailed some challenges and unanticipated outcomes.

## Background

Most people prefer treatment in the community, rather than as an inpatient. The Australian Clinical Guidelines for Stroke Management [1] and Rehabilitation Stroke Services Framework [2] both recommend that early supported discharge (ESD) models of care should be standard care for survivors of mild to moderate stroke. ESD is "a model that links inpatient care with community services with the aim of reducing length of stay", with therapy provided in the home in some variants of the model [1]. Previous recommendations state that ESD should form part of the stroke care continuum along with specialist stroke units [3, 4], as these models have been found to reduce long term dependency and length of hospital stay [5]. A good quality evidence base supporting ESD has enabled its implementation internationally over the past decade.

However, currently available research mostly focuses on patient outcomes and experiences, with little known about ESD implementation into existing stroke care services. A critical review of the initial years of ESD highlighted that intervention characteristics, sociocultural and other environmental aspects, and the ways in which this model of care is implemented varies widely [3]. While guidelines are available for ESD team composition, teamwork practices and the intervention itself, considerably diversity in implementation has become apparent [4].

Previous studies of ESD implementation from the staff perspective have generally reported positive perceptions of its success and translatability into practice. Adaptability, rehabilitation assistant input and cross-service working arrangements were highlighted as key facilitators, while a lack of referral clarity, social care delays and a dearth of appropriate post discharge services and resources were significant barriers [6,7]. Two Australian studies also highlighted potential differences in perceptions between stakeholder groups. While patient perceptions were generally positive, staff were only moderately favourable as they doubted ESD could provide hospital equivalent therapy intensity[8, 9].

The case study described here addresses a clear gap in understanding ESD implementation, by iteratively seeking referrer and staff perceptions over a six-month period. This case study also utilised an embedded implementation science theoretical framework, to rigorously understand determinants that influenced patient outcomes during a trial of an ESD model of care.

### Aim

To describe staff perceptions of the trial of an ESD model of care for stroke survivors at a large metropolitan public hospital in Australia.

### Materials and methods

This study utilised a mixed methods case study design, to describe the perspectives of staff working in a health service systematically trialling a new intervention [10]. The case was the

trial implementation of ESD during 2017, with ethics approval was received from the health organisations Human Research Ethics Committee (QA2017.14). Case studies provide an in-depth analysis of an event, program, or process, and are bounded by time and activity [11]. A researcher with expertise in knowledge translation (DH) used a partially embedded implementation research approach to closely collaborate with co-located health staff [12]. The study setting was a public health organisation located in a major Australian city. This organisation delivers acute tertiary, subacute, specialist ambulatory, and community-based services to a community of approximately 800,000 people. Service locations including three acute hospital campuses, a day hospital and a transition care program, and the health workforce numbers approximately 6,500 staff.

Prior to 2017, this organisation did not offer ESD to stroke survivors. A gap analysis revealed these patients often faced significant delays before receiving Community Based Rehabilitation (CBR) following discharge, and were only eligible for once weekly physiotherapy during post-acute care for up to 30 days while waiting. The organisation recognised they were not meeting best practice stroke care guidelines, and received funding to trial an ESD model of care over twelve months. The patient, carer and service outcomes of this trial are published elsewhere [13].

ESD models of care share common features and components, including multidisciplinary team members, a workforce with specialist stroke care knowledge and the provision of services in the community [14, 15]. However, ESD is has been implemented variably around the world, particularly in regards to recommended full time staff loadings and intensive treatment duration.

In this study, ESD was offered as part of services provided by a community rehabilitation team supported by an ESD coordinator. Patients were referred by the inpatient rehabilitation ward if they were medically stable post stroke, considered suitable for safe home discharge, able to be treated in the home environment and requiring intensive rehabilitation from at least two disciplines. ESD care was delivered by a multidisciplinary team including allied health clinicians, nurses, pharmacists, general practitioners and a rehabilitation consultant. Care was provide for four weeks (up to five days per week), with session frequency determined by individual patient needs. All patients with ongoing rehabilitation needs at the time of ESD discharge were referred on to community based rehabilitation services. Accessible and specific information about the ESD model and pathway of care was provided to all stroke survivors and carers, and the ESD team met on a weekly basis to coordinate care provision.

### Participants

Participants for this study were selected purposively [16], and there were two groups–staff who referred patients for ESD (Referrers), and staff involved in the planning, implementation or delivery of ESD during the trial (Delivering Staff). All health service staff meeting these criteria was invited to participate at each time point, and could participate in all, some or none of the data collection.

### Data collection

The study design was informed by the Consolidated Framework for Implementation Research (CFIR) [17]. The CFIR describes constructs identified from previous research as influential on effective knowledge translation. The framework supports analysis of the relationships between constructs and implementation outcomes [18], and can be used prospectively or retrospectively. The CFIR has five domains (intervention characteristics, outer setting, inner setting, characteristics of individuals and process), and 26 embedded constructs. Detailed definitions

for each domain and construct have been developed [19, 20], and are described below in Fig 1 in specific reference to ESD.

All the domains are interdependent, and constructs interact throughout implementation processes at the individual, service, organisational and community levels [17]. While complexity can be perceived as problematic to evaluation design, implementation is founded upon multiple influences, environments and interpersonal relationships [21]. All CFIR domains (but not all constructs) were addressed during the collection of both quantitative and qualitative data. Data related to trialability was not directly sought, as the trial as a whole tested ESD in this setting. Data about other personal attributes cannot be specifically collected given the lack of specific attributes in this construct, although relevant findings did emerge. Planning for this trial of ESD was completed prior to this study, and it was anticipated this data would be present in responses provided in relation to other CFIR domains and constructs. A matrix displaying the alignment between data collection and the CFIR is available in Supplementary Materials (S1 Appendix).

Two data collection approaches were used–mixed methods surveys and qualitative semi-structured interviews and/or focus groups. Participants had the option of participating in interviews or focus groups to support choice, and reduce burden associated with either format. The surveys were undertaken with both Referrers and Delivering Staff at three time points ($T_0$, $T_1$, $T_2$), while focus groups and interviews were conducted with Delivering Staff only at the final time point ($T_2$). Informed consent for survey data was assumed if responses were returned, however specific written consent was obtained for all focus groups and interviews.

## Outcome measures

A bespoke mixed methods survey was designed for each participant group based upon CFIR Interview Guide Tool [22], to support fidelity to the framework. Qualitative questions from this tool were converted to Likert items, and each survey also included space for open comments (see S2 Appendix). All data was collected anonymously (with only workforce group identified to enable sample description), and the survey took around five minutes to complete. Anonymity was preserved due to ethical concerns around inadvertent identification given the case study method, and also accommodated regular workforce turnover at the organisation. Qualitative prompts for the interviews and focus groups were also developed from the CFIR Interview Guide Tool [22]. These data collection sessions took 45–60 minutes, with all data digitally recorded and transcribed verbatim. While identical questions were posed for many variables, slightly different wording was adopted to enhance relevance to each participant group.

## Data analysis

Quantitative data were analysed descriptively, with tests of normality indicating that non-parametric statistics should be employed. Given the anonymous data, and that staff may not have participated at every time point, each group was considered independent for analysis. The Mann-Whitney U test was utilised the explore changes over time within each group, and also between the referring and delivering staff.

To measure changes over time, the trial was separated into two phases; Phase 1 encompassed baseline ($T_0$) to 3 months ($T_1$), while Phase 2 ran from 3 months ($T_1$) to the trials' conclusion after 6 months ($T_2$). Changes resulting from implementation occur over time, and implementation studies often only undertake data collection at baseline and post implementation to explore these changes [23]. However, the research team proposed the CFIR constructs could have differing levels of influence over the course of the ESD trial. For example;

## Intervention Characteristics

- **Intervention Source:** Origins of ESD
- **Evidence Strength & Quality:** Rigour of supporting evidence for ESD
- **Relative Advantage: B**enefits of ESD in comparison to standard care or alternatives
- **Adaptability:** Potential for ESD to be modified, changed or customised for local conditions
- **Trialability:** Ability to pilot ESD, before deciding to scale up or de-implement.
- **Complexity:** Perceived level of compliction associated with ESD duration, scope, intricacy, number of steps, and degree of difference
- **Design Quality & Packaging:** Format and presentation of supporting materials
- **Cost:** Economic impact of ESD implementation

## Outer Setting

- **Patient Needs & Resources:** Ability of ESD to meet patient needs
- **Cosmopolitanism:** Extent of network between health service and external organisations
- **Peer Pressure:** Competitive pressure to implement ESD following adoption by other health services
- **External Policy & Incentives:** Factors beyond the health service which either support or hinder ESD implementation

## Inner Setting

- **Structural Characteristics:** Health services age, size and organisational framework
- **Networks & Communications:** Social network within health service relevant to ESD
- **Culture:** Health service beliefs, norms & values
- **Implementation Climate:** Organisational context for ESD including tension for change, compatability with existing work, relative priority, incentives & rewards, goals & feedback and learning climate
- **Readiness for Implementation:** Indicators of ESD support including leadership engagement, available resources and access to knowledge & information

## Individual Characteristics

- **Knowledge & Beliefs:** Individual understanding & perceptions of ESD
- **Self-Efficacy:** Individual belief in capacity and capability for ESD
- **Individual Stage of Change:** Individual readiness to implement ESD
- **Individual Identification with Organisation:** Workforce perceptions of health service
- **Other Personal Attributes:** Other individual characteristics relevant to ESD

## Process

- **Planning:** Activities undertaken in advance of ESD implementation
- **Engaging:** Activities undertaken to encourage participation by key stakeholders, including opinion leaders, formally appointed internal implementation leaders, champions and external change agents
- **Executing:** Activities undertaken to achieving ESD implementation
- **Reflecting & Evaluating:** Feedback and reflection on the outcomes of the ESD implementation process

**Fig 1. Domains and constructs of the consolidated framework for implementation research [19, 20].**

perceptions regarding intervention source may be most relevant as implementation commences, when stakeholders were receiving their initial information about ESD. These nuances in the development of perceptions would be obscured by the use of a repeat measure tests (such as ANOVA), and hence the decision to analyse in two separate phases.

Qualitative survey data, and from interviews and focus groups, was analysed using a priori thematic analysis, [24] with the CFIR constructs as codes. A codebook was developed from the definitions for each domain and construct [19, 20] and including illustrative examples from the transcripts. Two researchers (DH, AM) independently assigned an interpreted meaning to each passage, on a line-by-line basis, for 25% of the transcripts. Very few instances of divergence existed (3.23% of total codes), and these were resolved via discussion between the researchers. A single researcher (DH) then assigned interpreted meanings to passages in all other transcripts. Two other researchers (KL, SN) independently reviewed these codes against the code book, and again low rates of divergence were found (KL 2.95%, SN 3.23%) and resolved by consensus.

Key relationships between CFIR domains and constructs were also analysed at the conclusion of qualitative analysis. The code co-occurrence function of the Dedoose platform was used to describe relationships between these concepts, which assigns frequencies to codes assigned to overlapping excerpts [25]. These relationships were then displayed in a network graph, with constructs related to $\geq 3$ other constructs identified as being key to the staff experience. All data sources for each CFIR domain and construct were then integrated in the final, mixed methods analysis. Within each construct, instances of consonance and dissonance between the data sources were described, and finally synthesised at the domain level.

## Results

A total of 111 surveys were received, and 23 staff participated in focus groups or interviews. Table 1 confirms participants came from a range of workforce groups.

### Characteristics of Individuals

As shown below in Table 2, participants perceived their personal characteristics and attributes in relation to ESD fairly positively throughout the trial. Greater improvements in overall perceptions occur in both groups during Phase 1, however increases in overall knowledge occurred during different phases for each participant group.

**Table 1. Participant characteristics.**

| Data Collection | Delivering Staff | Referrers |
|---|---|---|
| $T_0$ Survey | n = 20: Admin. (n = 3), AHA (n = 2), Nursing (n = 1), OT (n = 5), Physio. (n = 5), Psych. (n = 1), ST (n = 2) | n = 26: Medical (n = 1), Neuropsych. (n = 1), Nursing (n = 6), OT (n = 7), Physio. (n = 6), ST (n = 3), SW (n = 2), Unspecified (n = 1) |
| $T_1$ Survey | n = 14: Admin. (n = 2), OT (n = 4), Physio. (n = 4), Psych. (n = 1), ST (n = 1) | n = 18: Neuropsych. (n = 1), Nursing (n = 1), OT (n = 6), Physio. (n = 4), ST (n = 2), SW (n = 1) |
| $T_2$ Survey | n = 14: Nursing (n = 1), OT (n = 4), Physio. (n = 2), ST (n = 4), SW (n = 1) | n = 19: Nursing (n = 1), OT (n = 7), Physio. (n = 5), ST (n = 3), SW (n = 1) |
| Interviews / Focus Groups | n = 23: | |
| | Interviews (n = 7): Participants included Medical, ST, Neuropsych., OT and Physio. | |
| | Focus Groups (n = 16): Participants included Admin., Nursing, ST, Psych., OT and Physio. | |

In the following discussion, abbreviations denote the source of each quote (i.e. SI = staff interview, FG = focus group, RS = referrer survey), and delivering staff will be referred to simply as staff.

**Table 2. Findings for characteristics of individuals domain.**

| Construct & Key Quotes | Referrers (Scale 0–4) | |
|---|---|---|
| | Phase 1 | Phase 2 |
| **Overall Knowledge** | $T_0 = 3.10$, $T_1 = 3.06$ | $T_1 = 3.06$, $T_2 = 3.43$ |
| | $U = 175.0$, $p = 0.90$ | $U = 89.5$, $p = 0.17$ |
| **Overall Perception** | $T_0 = 3.25$, $T_1 = 3.50$ | $T_1 = 3.50$, $T_2 = 3.57$ |
| | $U = 134.50$, $p = 0.19$ | $U = 125.00$, $p = 0.98$ |
| | **Delivering Staff (Scale 0–4)** | |
| | Phase 1 | Phase 2 |
| **Overall Knowledge** *"The obvious advantage of ESD is reducing length of stay on both acute and subacute inpatient units"* (SI 5) | $T_0 = 2.96$, $T_1 = 3.32$ | $T_1 = 3.32$, $T_2 = 3.33$ |
| | $U = 183.50$, $p = 0.17$ | $U = 161.50$, $p = 1.00$ |
| **Overall Perception** *"it's getting to people in an earlier part of their journey, when they are going to be making more spontaneous neuroplastic changes"* (SI 3) | $T_0 = 3.33$, $T_1 = 3.63$ | $T_1 = 3.63$, $T_2 = 3.72$ |
| *"You're seeing more of the things that you can actually do that will make a difference to that person's life, rather than just the medical view".* (FG 1) | $U = 183.00$, $p = 0.10$ | $U = 162.50$, $p = 0.81$ |
| **Self Efficacy** *I know what intensive therapy looks like . . . .. (but) they had to shift their thinking, because these clients were going to achieve their goals much quicker"* (SI 7) | $T_0 = 3.05$, $T_1 = 3.39$ | $T_1 = 3.39$, $T_2 = 3.54$ |
| *"I think that it certainly does add to the workload, both ours and our inpatient colleagues"* (SI 1) | $U = 103.50$, $p = 0.03^*$ | $U = 123.00$, $p = 0.93$ |
| **Colleague Efficiency** | $T_0 = 3.05$, $T_1 = 3.39$ | $T_1 = 3.39$, $T_2 = 3.54$ |
| *"I actually started at [health organisation] halfway through this trial . . . it wasn't until I had an ESD patient that they were, oh, you need to prioritise this patient . . . I didn't know even what ESD stood for!"* (FG 1) | $U = 129.00$, $p = 0.14$ | $U = 103.00$, $p = 0.59$ |
| **Individual Identification with Organisation** *"You need a really robust inter-professional team, we have those people already here"* (SI 6) | $T_0 = 3.40$, $T_1 = 3.59$ | $T_1 = 3.59$, $T_2 = 3.71$ |
| *"We were ready to do something innovative and different"* (SI 1) | $U = 130.00$, $p = 0.23$ | $U = 109.00$, $p = 0.79$ |

$T_0$ = Baseline, $T_1$ = Three Months, $T_2$ = Six Months

$^*$ = Statistically Significant, $\alpha \leq 0.05$

Staff generally perceived ESD as closely aligned to their personal and professional beliefs about best practice and early intervention. Providing rehabilitation at home was also identified as a key aspect of ESD, which enabled meaningful goal setting and client centred practice. However, ESD implementation was both congruent with, and challenging to, their existing rehabilitation practice knowledge. Referrers noted that Grade 1 staff required increased support in Phase 1 to develop self-efficacy, while staff noted ESD knowledge was not consistently developed for staff members joining the organisation mid-trial.

A key tension identified was a belief that offering ESD to the trials' treatment group provided those patients with an unfair advantage. An unintended consequence of these beliefs was an emphasis on the shortcomings of standard practices, which remained in place for most patients. Diverse beliefs around the time commitment required by ESD were also evident, with

consistent (but not universal) claims of increased workload made throughout this study. Staff reported generally positive organisational perceptions, which supported a sense of commitment, enthusiasm and pride specifically associated with the ESD trial.

## Intervention characteristics

Participants also maintained generally positive perceptions about ESD itself, as shown below in Table 3. A mixed profile of changes was found, with some constructs experiencing only moderately fluctuations and others changing more markedly in specific phases. Decreases in some constructs (such as number of steps, degree of difference and cost) represent positive responses, as these are constructs were less is better.

Participants were generally well aware of the origin of ESD and its supporting evidence, which was perceived to provide support for good patient and service outcomes. ESD was unequivocally perceived to have more advantages than other stroke rehabilitation programs (as indicated by the perceived disparity between ESD and standard practices). These advantages were perceived to come at no cost or disadvantage to patients, fulfilling both their and the service's needs.

Perceived adaptability of ESD was identified both within the intervention, and as a function of the deployment of available resources (an Inner Setting construct). Perceptions of the duration and scope of ESD also became more positive, with duration influenced at times by staff attempting to meet their commitment to client centred practice. Perceptions of scope increased particularly in Phase 2 as staff developed more advanced skills in the ESD model, while perceptions about complexity were expressed in a relative sense, in relation to other interventions and systems.

While quantitative data indicated decreased costs perceived over time, some participants expressed cynicism about the implementation of ESD as a primarily cost cutting measure within their qualitative responses. The trial funding did not include additional staffing, and so participants had supported its implementation within their usual duties, leading some to question if identified cost savings were 'real'.

## Outer setting

The most prevalent construct identified was patient needs and resources, which was expressed from two perspectives—the general needs of patients at this organisation, and the specific needs of patients and carers in the early stages of stroke recovery. Staff discussed the impact of poverty, disadvantage and migration experiences within the local community on ESD, nothing that the model of care supports the use of interpreters via advance booking of appointments. Participants also highlighted that stroke survivors were not the only patient group which could potentially have their needs met through ESD models of care.

A major patient need met by ESD was returning home, which was perceived to be the optimal recovery environment; "*They have to get dressed, they have to get up and make their own cup of tea, they have to–you know, so it's kind of forced rehab . . .*" (FG1). Staff reported that both patients and carers shared this perception, reacting positively to the prospect of ESD when initially approached. However, participants expressed concerns about ESD's ability to meet family needs in early recovery, particularly as patients are returning home with higher levels of dependence; "*That's the one piece of feedback I get from every single family member, I didn't realise how hard this was going to be and I didn't realise what it meant to be caring for them*" (FG 1). A possible response suggested by staff was extending the ESD model beyond patients to support family and carers, who were acknowledged as key stakeholders.

**Table 3. Findings for Intervention Characteristics domain.**

| Construct & Key Quotes | Delivering Staff (Scale 0–4) | |
| --- | --- | --- |
| | **Phase 1** | **Phase 2** |
| **Rationale for Development** | $T_0 = 3.65$, $T_1 = 3.67$ | $T_1 = 3.67$, $T_2 = 3.86$ |
| | $U = 170.50$, $p = 0.79$ | $U = 108.00$, $p = 0.51$ |
| **Who Developed ESD** | $T_0 = 2.90$, $T_1 = 2.94$ | $T_1 = 2.94$, $T_2 = 3.21$ |
| | $U = 172.00$, $p = 0.82$ | $U = 108.00$, $p = 0.51$ |
| **Evidence Strength and Quality** "*The literature sort of supported that it could be beneficial, and that there weren't going to be any adverse events, and the patients generally liked it*" (SI 2) | $T_0 = 3.00$, $T_1 = 3.06$ | $T_1 = 3.06$, $T_2 = 3.43$ |
| | $U = 108.50$, $p = 0.06$ | $U = 101.0$, $p = 0.35$ |
| **Relative Advantage–Other Alternatives** "*it reduces bed days. . . so, that reduces risk of infections . . . risk of pressure injuries . . . demand on the ED department, just by having someone being able to move out of that bed sooner has allowed for other things to happen within the hospital.*" (FG 1). | $T_0 = 2.32$, $T_1 = 3.06$ | $T_1 = 3.06$, $T_2 = 3.43$ |
| | $U = 108.50$, $p = 0.06$ | $U = 101.0$, $p = 0.35$ |
| **Relative Advantage–Similar Programs** "*we're not trying to convince them to go down an option that suits us but it doesn't suit them . . . like a double positive, like it seems to be financially better and actually better for the clients*" (FG 2) | $T_0 = 2.40$, $T_1 = 2.94$ | $T_1 = 2.94$, $T_2 = 3.36$ |
| | $U = 130.00$, $p = 0.15$ | $U = 97.00$, $p = 0.28$ |
| **Adaptability** "*allows a bit more flexibility to swap people between sites and respond to changes in wait lists*" (SI 6) | $T_0 = 2.55$, $T_1 = 3.28$ | $T_1 = 3.28$, $T_2 = 2.93$ |
| "*we kind of made it a bit more [health service] specific so that we could really target our audience*" (SI 2) | $U = 94.5$, $p = 0.01^*$ | $U = 92.5$, $p = 0.21$ |
| **Complexity–Duration** | $T_0 = 1.21$, $T_1 = 1.39$ | $T_1 = 1.39$, $T_2 = 1.64$ |
| | $U = 158.5$, $p = 0.72$ | $U = 104.50$, $p = 0.42$ |
| **Complexity–Scope** | $T_0 = 1.56$, $T_1 = 1.53$ | $T_1 = 1.53$, $T_2 = 2.07$ |
| "*the clinicians who've established their roles . . . have become more complex and that has impacted on the skill set in those roles*" (SI 3) | $U = 162.0$, $p = 0.98$ | $U = 85.50$, $p = 0.19$ |
| **Complexity–Intricacy** | $T_0 = 1.94$, $T_1 = 1.89$ | $T_1 = 1.89$, $T_2 = 2.29$ |
| | $U = 158.0$, $p = 0.91$ | $U = 99.00$, $p = 0.31$ |
| **Complexity–Number of Steps** | $T_0 = 1.95$, $T_1 = 1.94$ | $T_1 = 1.94$, $T_2 = 1.86$ |
| | $U = 163.0$, $p = 0.82$ | $U = 120.50$, $p = 0.85$ |
| **Complexity–Degree of Difference** | $T_0 = 2.26$, $T_1 = 1.83$ | $T_1 = 1.83$, $T_2 = 2.08$ |
| "*More in the context of the other things that were going on rather than what was happening within ESD*" (SI 1). | $U = 123.0$, $p = 0.15$ | $U = 101.50$, $p = 0.54$ |
| **Complexity–Quality of Supporting Materials** | $T_0 = 2.79$, $T_1 = 3.22$ | $T_1 = 3.22$, $T_2 = 3.14$ |
| | $U = 125.00$, $p = 0.17$ | $U = 116.0$, $p = 0.72$ |
| **Cost** | $T_0 = 2.61$, $T_1 = 2.06$ | $T_1 = 2.06$, $T_2 = 1.93$ |

(*Continued*)

**Table 3.** (Continued)

| Construct & Key Quotes | Delivering Staff (Scale 0–4) | |
|---|---|---|
| | Phase 1 | Phase 2 |
| "*the cost of providing our service is less than if they were an inpatient so that's good for the organisation . . . money kind of talks sometimes at higher levels more than other things*" (FG 2) | U = 116.50, p = 0.23 | U = 108.50, p = 0.69 |

$T_0$ = Baseline, $T_1$ = Three Months, $T_2$ = Six Months

* = Statistically Significant, $\alpha \leq 0.05$

## Inner setting

Perceptions relating to the Inner Setting domain were mixed, with Table 4 below showing both positive and negative changes over time.

Staff generally perceived ESD as aligning closely with organisational norms and values, particularly around the provision of best care and an organisational commitment to innovation. Perceptions of an innovative culture may also relate to the overall implementation climate; however, other aspects of this construct (goals and feedback, learning climate, organisational incentives and reward) had very limited presence in the data. The relative priority of ESD within the organisation was understood by staff to interact with competing priorities, however they perceived a strong tension for change.

Referrers reported more negative perceptions of ESD's impact on workload than staff, however qualitative responses indicated this was expected and was *"manageable given good planning and organisation"* (RS). ESD was not initially perceived as compatible with the CBR context, with several participants describing feeling forced to choose between models of care rather than adopting a hybrid approach. These choices manifested themselves in changes to long held practices, which were particularly challenging for some of the smaller professions

**Table 4. Findings for Inner setting domain.**

| Construct & Key Quotes | Referees (Scale 0–4) | |
|---|---|---|
| | Phase 1 | Phase 1 |
| **Compatibility with Existing Workflows** | $T_1$ = 2.88, $T_2$ = 2.47 | $T_1$ = 2.47, $T_2$ = 2.61 |
| | U = 177.00, p = 0.18 | U = 149.00, p = 0.69 |
| | Delivering Staff (Scale 0–4) | |
| **Compatibility with Existing Workflows** | $T_0$ = 2.53, $T_1$ = 3.18 | $T_0$ = 3.18, $T_1$ = 3.07 |
| "*They (referrers) are having to spend a lot more time dedicated on these potential ESD clients to get everything done. . . that increase in workload has been stressful*" (FG 1). | U = 108.50, p = 0.21 | U = 116.50, p = 0.94 |
| "*Because basically, you're trying to manage an inpatient caseload in an outpatient setting*" (FG 1) | | |
| **Access to Education and Information** | $T_0$ = 2.63, $T_1$ = 3.06 | $T_0$ = 3.06, $T_1$ = 3.07 |
| | U = 130.00, p = 0.22 | U = 124.00, p = 0.95 |

$T_0$ = Baseline, $T_1$ = Three Months, $T_2$ = Six Months

* = Statistically Significant, $\alpha \leq 0.05$

and non-clinical staff. While these changes were perceived as a positive opportunity to work in new ways by some, others found they challenged beliefs around the core business of CBR.

### Process

Overall, perceptions of the implementation process remained steady for both referrers and staff over time, as shown below in Table 5.

As expected, planning was not a strong theme in the data. However some staff reflected on the value of reviewing organisational data, workforce consultations and benchmarking against other services to inform the trial process. Attempts were also made to anticipate potential process and workflow issues, and differing perceptions between stakeholders, although this proved to be difficult without precedents and prior experience.

High levels of staff investment were consistently identified as important by participants, and additional investment provided by management and informal ESD leaders (such as team leaders, managers, the steering committee, nurse unit managers and nurse practitioners) was also recognised within the CBR service. This solid engagement was attributed both to the perceived alignment between ESD and best care, and workforce perceptions of being able to meaningfully influence implementation. While opinion leaders and external change agents were not discussed in this data, the ESD co-ordinator was consistently identified as a key champion. Perceptions of her role were universally positive, with accessibility, excellent clinical knowledge, face-to-face attendance of team meetings, an ability to work across service boundaries and a single point of contact and coordination highlighted as key factors contributing to its success.

The intensive nature of ESD continuously challenged its execution over time, with the ability to retain flexibility perceived as crucial by participants. The early stages of ESD execution were experienced as uncertain by some, however there was a sense the workforce could abide with it and understood uncertainty was a necessary part of the implementation process. By $T_2$, most participants expressed considerable satisfaction and confidence with the ESD trial at this organisation. Despite the challenges identified at previous time points, ESD was now perceived as *"business as usual and so we sort of know how it works and know what's happening"* (CI 1). However, not all staff were completely comfortable with ESD by this point, indicating six months was not sufficient time for everyone to fully adapt to the new intervention. By the trials conclusion, participants generally believed they had enough evidence to support its ongoing sustainability.

### Key relationships between constructs

As shown in Fig 2, there were multiple relationships between the different constructs in this study.

Key relationships are displayed below in Table 6, several of which span multiple domains. While patient needs and resources were a key motivator for staff, the other key constructs identified also had a significant impact on the feasibility and sustainability of ESD during the trial.

### Discussion

This study found participants generally perceived the model trialled to have a positive impact on care for stroke survivors. In particular, themes related to the value of flexibility, greater understanding of patient needs in the home environment, and ESD's potential for supporting quality rehabilitation align with previous international findings [6, 7]. However, other findings differed from previous research, which may reflect local contextual factors. Challenges with

**Table 5. Findings for process domain.**

| Construct & Key Quotes | | |
|---|---|---|
| | **Referees (Scale 0–4)** | |
| | **Phase 1** | **Phase 1** |
| **Champion Communication** | $T_1 = 3.78, T_2 = 3.58$ | $T_1 = 3.58, T_2 = 3.78$ |
| | $U = 99.50, p = 0.32$ | $U = 128.00, p = 0.29$ |
| **Ease of Referral** | $T_1 = 3.33, T_2 = 3.16$ | $T_1 = 3.16, T_2 = 3.78$ |
| "We were getting a lot of, what I would call softer clients, initially. But now . . . I think that they now trust the system and trust that we can do the job" (FG 1) | $U = 229.50, p = 0.76$ | $U = 83.50, p = 0.02^*$ |
| **Ease of Transfer** | $T_1 = 3.21, T_2 = 3.21$ | $T_1 = 3.21, T_2 = 3.72$ |
| "we decided we were not going to do the assessment in the format we would ordinarily do because we already have the information,. . . its always been like that. . . so that's why we've stuck with it . . . we should get rid of that" (CI 4) | $U = 202.0, p = 0.73$ | $U = 91.5, p = 0.04^*$ |
| **Identifying Suitable Patients** | $T_1 = 3.11, T_2 = 3.16$ | $T_1 = 3.16, T_2 = 3.44$ |
| | $U = 232.50, p = 0.81$ | $U = 124.50, p = 0.24$ |
| **Referral Satisfaction** | $T_1 = 3.42, T_2 = 3.21$ | $T_1 = 3.21, T_2 = 3.78$ |
| "We were just told to do stuff . . . and we were told it was a trial, so we didn't know how long the trial was going to be . . . but it's alright, we've adapted and we're moving on" (FG 1) | $U = 213.50, p = 0.81$ | $U = 90.50, p = 0.04^*$ |
| **Transfer Satisfaction** | $T_1 = 3.33, T_2 = 3.32$ | $T_1 = 3.32, T_2 = 3.61$ |
| "notable improvement in IP staff planning of D/Cs and flagging of pts for ESD which has decreased pressures" (RS) | $U = 204.00, p = 0.77$ | $U = 115.00, p = 0.21$ |
| "when you're starting something new you certainly don't have all the answers . . . just starting and ironing things out as they went . . . That's what I learnt through ESD, you had to just start, and then work it out as you go" (CI 7) | | |
| **Champion Satisfaction** | $T_1 = 3.81, T_2 = 3.58$ | $T_1 = 3.58, T_2 = 3.94$ |
| "I think if we didn't have [REDACTED] or a coordinator, it would have fell apart . . . it's important to have someone who is driving this process and that can be that main point of call to keep things moving along smoothly" (FG 2) | $U = 186.00, p = 0.19$ | $U = 110.50, p = 0.16$ |
| | **Delivering Staff (Scale 0–4)** | |
| | Phase 1 | Phase 1 |
| **ESD Satisfaction** | $T_0 = 3.44, T_1 = 3.50$ | $T_0 = 3.50, T_1 = 3.64$ |
| "We had to design a program that the ward staff were going to feel confident with, because they were not going to send their inpatient rehab patients home early if we weren't going to deliver" (CI 7) | $U = 141.00, p = 0.51$ | $U = 117.00, p = 0.75$ |
| **ESD Effectiveness** | $T_0 = 3.05, T_1 = 3.61$ | $T_0 = 3.61, T_1 = 3.64$ |
| "The ESD program has been absolutely fantastic, and appears to have made a huge contribution to improving patient" (RS) | $U = 96.5, p = 0.02^*$ | $U = 121.00, p = 0.87$ |

$T_0$ = Baseline, $T_1$ = Three Months, $T_2$ = Six Months

$^*$ = Statistically Significant, $\alpha \leq 0.05$

onward referral or liaison with other services reported by the previous study [6] were not identified here, possibly due to differences in the Outer Setting between nations. Previous Australian research expressed a similarly critical stance towards this service model [8, 9], however

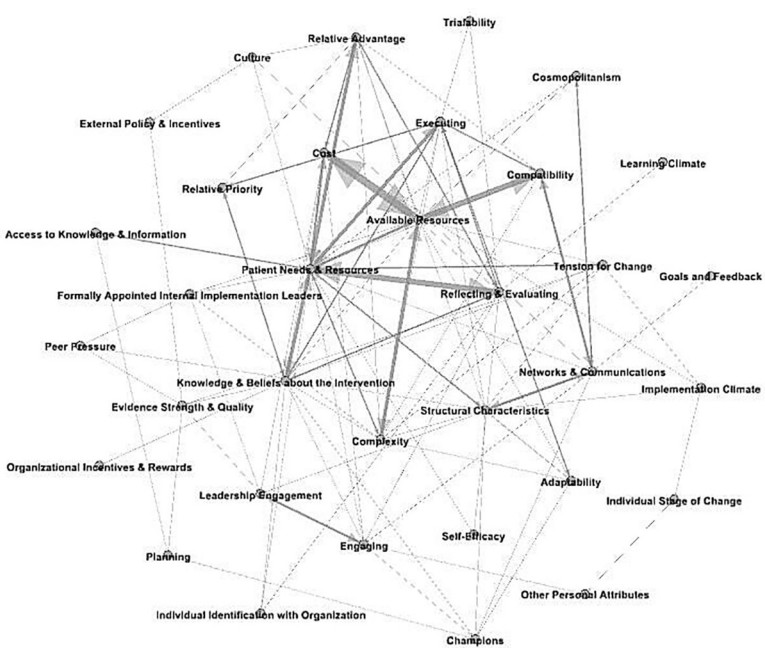

**Fig 2. Key relationships between domains and constructs.**

participants in this study were generally more positive towards ESD than participants in the previous studies. These areas of agreement and difference may tentatively indicate which aspects of the ESD model are core (and transferable as essential elements), and which are peripheral (and therefore adaptable to local conditions) [17]. Given the variability in ESD models of care being implemented globally [4, 15], further research to describe its core and peripheral characteristics is important to guide its effective use.

Beyond findings specific to the implementation of ESD models of care, alignment with the CFIR enabled the research team to focus on the determinants of implementation more generally. The research team notes the CFIR construct of complexity [17] includes several elements which are proxies for this concept (i.e. perceived difficulty, duration, scope, radicalness and disruptiveness), but not the key element of relationships between components [26]. Participant perceptions of the positive outcomes of ESD were supported by data related to several

**Table 6. Key constructs Identified by staff and their relationships with other constructs.**

| Key Construct (Domain) | Related Constructs (Domain) |
|---|---|
| Patient Needs and Resources (Outer Setting) | Knowledge and Beliefs about ESD (Characteristics of Individuals) |
| | Relative Advantage (Intervention Characteristics) |
| | Executing (Process) |
| | Reflecting (Process) |
| Networks and Communications (Inner Setting) | Compatibility (Inner Setting) |
| | Structural Characteristics (Inner Setting) |
| Available Resources (Inner Setting) | Compatibility (Inner Setting) |
| | Complexity (Intervention Characteristics) |
| | Cost (Intervention Characteristics) |
| | Patient Needs & Resources (Outer Setting) |
| Leadership (Inner Setting) | Engaging (Process) |

constructs previously identified as being supportive of implementation success. ESD was trialled by a highly skilled workforce, who already possessed good self perceived knowledge and confidence at baseline. While good knowledge and positive beliefs are not guarantees of effective implementation, they are important psychological and reflective influences on practice change [27]. The identified champion for trialling ESD, who was also intervention coordinator, also clearly played a key role in supporting the workforce throughout the process. While Braithwaite et al. [28] suggests champions are more effective in implementing technological change, Wurtze et al. found (as did this study) they may also have an effective role in supporting behavioural change in complex health care systems [29].

ESD was also perceived to align strongly with personal and organisational values, particularly in regards to client centred care and equity. While the CFIR includes the impact of values on implementation as part of 'other personal attributes', other implementation models (such as the organisational theory of innovation [30]) foreground the fit between an innovation and values more strongly. Externally to the organisation, the release of revised National Clinical Guidelines for Stroke Management [1] halfway through this trial also reinforced the quality of evidence supporting ESD and reconfirmed for participants they were adopting best practice. The timing of when this evidence became available was an additional factor, which added to the motivation provided by values based perceptions of the model of practice.

However, participant perceptions of ESD were not uniformly positive, with some specific areas of critique or ambivalence identified. Increasing workload demands on already busy staff is a well-recognised barrier to practice change, and the different attitudes towards the validity of such perceptions may contribute to conflict which negatively effects workplace health and wellbeing [31]. However, this issue did appear to become less problematic for staff over time, as ESD transitioned into usual practice, suggesting that it is a more urgent issue in the earlier phases of implementation. Implementation of multidimensional interventions within complex adaptive systems can also result in unintended consequences, due to the intense interdependence between system components which limits our ability to predict emergent properties [32]. As noted by Brainard and Hunter [33], these consequences are often omitted from study reporting but the should be sought during the data collection phase to promote understanding. Their inclusion in the findings of this study flags the potential issues teams seeking to implement ESD may encounter, which can contribute to their planning.

A significant feature of this study was the various changes in perceptions of CFIR constructs over time, as the ESD model of care was adapted and further developed within the service. A longitudinal qualitative study of the implementation of values based health care in a psychiatric system [34] also found their intervention was adapted or changed over time, suggesting a less linear relationship between the planning and execution constructs than suggested by the CFIR Framework. Longitudinal methods in implementation research are currently in the minority, with less than half the studies in a recent review of CFIR application conducting sustained data collection [35]. Many current CFIR studies take a descriptive approach to the constructs, rather than exploring key relationships between them. The components of the CIFR are intended to interact at multiple levels, and in sophisticated and complex ways [17]. There are a range of approaches available to investigate how the framework behaves across dimensions, which would provide greater depth to future studies.

The impact of interactions between the domains and constructs of the CFIR on implementation is further highlighted by comparison of the findings of this study with another case study based in child psychiatry [36]. Barwick et al quantitatively measured the perceived importance of each construct, highlighting those which predominated in the experience of their sample. In common with this study, structural characteristics; networks and communication; and knowledge and beliefs about the intervention were particularly salient. However,

other constructs identified as important in this study (including patient needs and resources, cost, available resources and relative advantage) were not prominent in their findings. Unique patterns of salient CFIR features have also been found in other practice settings including psychiatry [34], the provision of online health information and referrals [37] and even between study sites in a study of veterans health primary care services on American Indian reservations [38]. This mitigates against finding the most commonly reported CFIR constructs that influence implementation, which has been suggested as a potential avenue for exploration [36].

### Limitations

The main limitation of this case study is its location within a single health service, and relatively small geographical area. As with all research using a case study approach, the findings presented here cannot be generalised easily to other health services, and could not be replicated even in the original setting due to the translation of its findings back into practice. However, the methodological strategies adopted to increase the rigour of this study (including the adoption of multiple methods, trustworthiness measures and inclusive recruitment) have ensured this is a robust example of this type of study. The amount of data collected, and multidimensional analysis undertaken, also drew on considerable resources leading to an extended period of time required to complete the study. Finally, the adoption of the CFIR as the embedded implementation model also means the limitations of this framework (including its scope as a determinant model [39], the number of constructs [40] and previously mentioned comments on the construct of complexity) are also inherent within this case study.

### Conclusions

This case study provides a detailed and multidimensional understanding of staff perceptions of the implementation of an ESD model of care for stroke survivors at a large metropolitan public hospital in Australia. Presenting the findings comprehensively in a single case study clearly highlights the sophistication of knowledge translation and implementation in modern complex healthcare environments. The domains and constructs of the CFIR were constantly interacting with each other throughout the case study, with interactions identified at individual, team and organisational levels across all the time points measured. From these interactions emerged a generally positive and effective implementation process, although it was not without its challenges and unanticipated outcomes.

The importance of understanding how ESD is implemented from multiple perspectives has been confirmed with a recent protocol for a realist evaluation study [41], which will utilised the iPARIHS framework. However, the real value of this case study has been the depth of understanding provided to the health service about what works well (and what doesn't work so well) in its local context. The knowledge gained from this case study has been applying directly back into ongoing implementation projects at the site, and contributed to an increased awareness of the necessity to address implementation alongside effectiveness.

### Supporting information

**S1 Data.**
(DOCX)

**S2 Data.**
(DOCX)

## Acknowledgments

This study was also supported by significant in kind contributions from the healthcare organisation, and the generosity of all participants in providing their insights and time.

## Author Contributions

**Conceptualization:** Danielle Hitch.

**Data curation:** Danielle Hitch, Kathleen Leech, Sharon Neale.

**Formal analysis:** Danielle Hitch, Kathleen Leech, Sharon Neale.

**Investigation:** Kathleen Leech, Sharon Neale, Avetta Malcolm.

**Methodology:** Danielle Hitch.

**Project administration:** Danielle Hitch, Kathleen Leech, Sharon Neale.

**Resources:** Danielle Hitch, Kathleen Leech, Sharon Neale.

**Supervision:** Danielle Hitch.

**Validation:** Danielle Hitch, Kathleen Leech, Sharon Neale, Avetta Malcolm.

**Writing – original draft:** Danielle Hitch, Kathleen Leech, Sharon Neale.

**Writing – review & editing:** Danielle Hitch, Kathleen Leech, Sharon Neale, Avetta Malcolm.

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
