## [Decision Letter · Decision Letter 0]

30 Oct 2019

PONE-D-19-26093

Evaluating the implementation of an Early Supported Discharge (ESD) program for stroke survivors: A mixed methods longitudinal case study

PLOS ONE

Dear Dr Hitch,

Thank you for submitting your manuscript to PLOS ONE. After careful consideration, we feel that it has merit but does not fully meet PLOS ONE’s publication criteria as it currently stands. Therefore, we invite you to submit a revised version of the manuscript that addresses the points raised during the review process.

We would appreciate receiving your revised manuscript by Dec 14 2019 11:59PM. To enhance the reproducibility of your results, we recommend that if applicable you deposit your laboratory protocols in protocols.io, where a protocol can be assigned its own identifier (DOI) such that it can be cited independently in the future. For instructions see: http://journals.plos.org/plosone/s/submission-guidelines#loc-laboratory-protocols

We look forward to receiving your revised manuscript.

Kind regards,

Maw Pin Tan, M.D.

Academic Editor

PLOS ONE

Journal Requirements:

Funding for the implementation of the ESD model of care at this healthcare organisation was provided by the Victorian Stroke Clinical Network. This study was also supported by significant in kind contributions from the healthcare organisation, and the generosity of all participants in

providing their insights and time.

Reviewers' comments:

Reviewer's Responses to Questions

**Comments to the Author**

1. Is the manuscript technically sound, and do the data support the conclusions?

Reviewer #1: Partly

Reviewer #2: No

2. Has the statistical analysis been performed appropriately and rigorously? 

Reviewer #1: Yes

Reviewer #2: No

3. Have the authors made all data underlying the findings in their manuscript fully available?

Reviewer #1: Yes

Reviewer #2: Yes

4. Is the manuscript presented in an intelligible fashion and written in standard English?

Reviewer #1: No

Reviewer #2: Yes

5. Review Comments to the Author

Reviewer #1: Recommend cutting the word count by half or more.

Introduction

The ESD program that has been implemented is never described. These programs vary a tremendous amount. There is no standard program at this time. Without an explanation of the program the reader has no context for understanding the results.

Material and methods:

Line 89 …explore the complexities-unclear Do you mean the staff and referrer perspective? There is an overuse of the word “complexities” in the manuscript which actually makes understanding the writing more complex. Simply state what you mean.

Line 95 …”a partially embedded implementation research partnership” –unclear.

Line 113 cite your other publication

Participants

Line 118-9 define who these people are. For instance, “staff” in this section is all who are involved in planning, implementing and delivery. However, in the tables staff only refers to “delivering staff”. It is very difficult to understand who was involved in what aspects of the study throughout this manuscript. Further clarify exactly who was took which surveys and who was involved in the focus groups.

Was any other data collected regarding demographics: Age, race, years in the profession, years employed at that hospital? Were they all stroke experts?

Data Collection

This entire section is unclear. The use of CFIR is a positive aspect of this study, but the explanation of the framework and how it is used is poorly executed. The reviewer had to look up the framework and read someone else’s explanation for clarification.

It seems like both one-on-one interviews and focus groups were conducted. How many occurred and what was the make-up of each? Was it semi-structured? What method was used to collect the data and what questions were asked or, at least, what constructs were covered?

It would be helpful to include the surveys as a supplement.

Outcome Measures

Line 157: What do the authors mean by “mixed method survey” Likert and open-ended questions? This is unclear. What are the participant groups?...referrers and staff?

Results

This section needs to be made much more concise. Also, it is difficult to understand the tables without referring to the supplemental document. The tables could be made clearer with the additional explanation. For example, Table 3 “complexity-duration” is this duration of the ESD program? What aspect? “Complexity-degree of difference”—the reader has no context for this. What does this mean?

Consider making the tables clearer and then incorporating a few of the quotes into the tables. This would significantly cut down on the word count in this section. The tables seem to tell most of the story, the pages upon pages of words are extraneous. At the end of the section the authors highlight ‘key constructs’. It is unclear how these were derived. However, the authors might consider explaining how they came up with these key concepts and then focus on these areas (using words) in the results section. Let the tables tell the story of the data for all the other areas.

Discussion

Line 612: What is the champion?

There is an entire paragraph written about the perception of patients ‘missing out’—what does this mean? Again, there is no context for the reader to understand this notion.

Consider linking the topics in the discussion with some transition sentences to improve the flow. This section reads like an extension of the results section.

Consider removing the paragraph lines 657-680. There is no need to continue a defense of the study methodology (use of CFIR) here. This was all covered earlier in the paper (although it needs to be clarified).

Reviewer #2: I have provided a detailed review in the attached report. I would like to thank the authors for reporting the findings from this study – there has been a lack of implementation of Early Supported Discharge services in Australia and so this will be of interest to the stroke community. However, I do think the paper needs considerably more work to get it to the standard required for publication and to ensure a reader would be able to interpret the findings correctly.

6. PLOS authors have the option to publish the peer review history of their article (what does this mean?). If published, this will include your full peer review and any attached files.

Reviewer #1: No

Reviewer #2: No

---

## [Author Response · Author response to Decision Letter 0]

17 Dec 2019

These responses have been provided in the attached cover letter, with a point by point address to each comment and recommendation

---

## [Decision Letter · Decision Letter 1]

2 Mar 2020

PONE-D-19-26093R1

Evaluating the implementation of an Early Supported Discharge (ESD) program for stroke survivors: A mixed methods longitudinal case study

PLOS ONE

Dear Dr Hitch,

Thank you for submitting your manuscript to PLOS ONE. After careful consideration, we feel that it has merit but does not fully meet PLOS ONE’s publication criteria as it currently stands. Therefore, we invite you to submit a revised version of the manuscript that addresses the points raised during the review process.

We would appreciate receiving your revised manuscript by Apr 16 2020 11:59PM. To enhance the reproducibility of your results, we recommend that if applicable you deposit your laboratory protocols in protocols.io, where a protocol can be assigned its own identifier (DOI) such that it can be cited independently in the future. For instructions see: http://journals.plos.org/plosone/s/submission-guidelines#loc-laboratory-protocols

We look forward to receiving your revised manuscript.

Kind regards,

Maw Pin Tan, M.D.

Academic Editor

PLOS ONE

Reviewers' comments:

Reviewer's Responses to Questions

**Comments to the Author**

1. If the authors have adequately addressed your comments raised in a previous round of review and you feel that this manuscript is now acceptable for publication, you may indicate that here to bypass the “Comments to the Author” section, enter your conflict of interest statement in the “Confidential to Editor” section, and submit your "Accept" recommendation.

Reviewer #1: (No Response)

Reviewer #2: All comments have been addressed

2. Is the manuscript technically sound, and do the data support the conclusions?

Reviewer #1: Partly

Reviewer #2: Yes

3. Has the statistical analysis been performed appropriately and rigorously? 

Reviewer #1: Yes

Reviewer #2: Yes

4. Have the authors made all data underlying the findings in their manuscript fully available?

Reviewer #1: Yes

Reviewer #2: Yes

5. Is the manuscript presented in an intelligible fashion and written in standard English?

Reviewer #1: No

Reviewer #2: No

6. Review Comments to the Author

Reviewer #1: The authors state that they feel that they are unable to substantially reduce the word count because they would lose "essential information". Yet the excessive text actually takes away from this paper. The reader is completely bogged down in the minutia and all the quotes to the point where one simply loses interest. The results of this study simply do not warrant over 7,000 words of text. The intro can be synthesized, the reader does not need to read a long paragraph describing previous work study by study. Secondly, this reviewer asked for the authors to include a description of their specific ESD program. The reader still does not know how the referral process works, who delivered that care in the home, for how long was care provided, was the care daily or weekly, how were patients selected for ESD, what the discharge process from ESD, was there continuing care after discharge from ESD? Without an in depth understanding of the program, this data is meaningless. The reader doesn't know anything about the service that the staff is being surveyed about. It truly renders these results meaningless. The tables are a nice addition and serve to clarify some. The reviewer, once again, suggests using the table to in the results section and highlighting the significant findings with text.

Reviewer #2: (No Response)

7. PLOS authors have the option to publish the peer review history of their article (what does this mean?). If published, this will include your full peer review and any attached files.

Reviewer #1: No

Reviewer #2: Yes: Rebecca Fisher

---

## [Author Response · Author response to Decision Letter 1]

16 Apr 2020

Please see attached detailed response to reviewers

---

## [Decision Letter · Decision Letter 2]

22 May 2020

PONE-D-19-26093R2

Evaluating the implementation of an Early Supported Discharge (ESD) program for stroke survivors: A mixed methods longitudinal case study

PLOS ONE

Dear Dr. Hitch,

Thank you for submitting your manuscript to PLOS ONE. After careful consideration, we feel that it has merit but does not fully meet PLOS ONE’s publication criteria as it currently stands. Therefore, we invite you to submit a revised version of the manuscript that addresses the points raised during the review process.

We look forward to receiving your revised manuscript.

Kind regards,

Maw Pin Tan, M.D.

Academic Editor

PLOS ONE

Reviewers' comments:

Reviewer's Responses to Questions

**Comments to the Author**

1. If the authors have adequately addressed your comments raised in a previous round of review and you feel that this manuscript is now acceptable for publication, you may indicate that here to bypass the “Comments to the Author” section, enter your conflict of interest statement in the “Confidential to Editor” section, and submit your "Accept" recommendation.

Reviewer #1: (No Response)

2. Is the manuscript technically sound, and do the data support the conclusions?

Reviewer #1: Yes

3. Has the statistical analysis been performed appropriately and rigorously? 

Reviewer #1: Yes

4. Have the authors made all data underlying the findings in their manuscript fully available?

Reviewer #1: Yes

5. Is the manuscript presented in an intelligible fashion and written in standard English?

Reviewer #1: Yes

6. Review Comments to the Author

Reviewer #1: This manuscript is much improved. It remains over 7,000 words (far too lengthy for a survey-based case study). I don't know what the word limit is for PLOS ONE for this type of paper.

I would suggest the following edits as well...

Line 61 Unclear, suggest…the intervention itself, when and how ESD is implemented varies widely.

Line 67-68 Unclear, suggest…staff perceptions were only moderately favorable, as they doubted ESD could provide hospital equivalent therapy intensity…

141Unclear. …and it was anticipated that questions about the intervention characteristics would elicit this information?? Unsure what the authors are trying to say here. Maybe :… and survey question development and data collection techniques were based on the CFIR framework. (??)

360-362 Recommend removing this sentence, “this study has also addressed several previously identified gaps in EDS implementation research, by including medical participants and survivors or severe stroke, and exploring co-ordination between hospital and community care.” The study does neither of these things. (1) stroke survivors are not participants in this study. The ESD program may include patients with severe stroke but the study is only about the perceptions of the staff regarding the ESD program. (2) the study does not explore coordination between hospital and community care. It only explores perceptions about coordination.

Figure 2 is very blurry

7. PLOS authors have the option to publish the peer review history of their article (what does this mean?). If published, this will include your full peer review and any attached files.

Reviewer #1: No

---

## [Author Response · Author response to Decision Letter 2]

27 May 2020

Response to Reviewers

This manuscript is much improved. It remains over 7,000 words (far too lengthy for a survey-based case study). I don't know what the word limit is for PLOS ONE for this type of paper.

 * We acknowledge the reviewer has consistently asked for the word length on this article to be reduced. During the previous two reviews we have reduced the word count by 25%, and we note that the other reviewer for this manuscript was satisfied with the current length of the article. The submission guidelines for this journal stipulate there are no restrictions on word count, number of figures or amount of supporting information. We disagree with your description of the study as a survey based case study – both forms of mixed methods data collected (survey and focus groups) have contributed significantly to the findings in this paper. 

Line 61 Unclear, suggest…the intervention itself, when and how ESD is implemented varies widely.

* Sentence revised as recommended 

Line 67-68 Unclear, suggest…staff perceptions were only moderately favorable, as they doubted ESD could provide hospital equivalent therapy intensity…

* Sentence revised as recommended

141Unclear. …and it was anticipated that questions about the intervention characteristics would elicit this information?? Unsure what the authors are trying to say here. Maybe :… and survey question development and data collection techniques were based on the CFIR framework. (??)

* Sentence revised with different wording to clarify meaning. 

360-362 Recommend removing this sentence, “this study has also addressed several previously identified gaps in EDS implementation research, by including medical participants and survivors or severe stroke, and exploring co-ordination between hospital and community care.” The study does neither of these things. (1) stroke survivors are not participants in this study. The ESD program may include patients with severe stroke but the study is only about the perceptions of the staff regarding the ESD program. (2) the study does not explore coordination between hospital and community care. It only explores perceptions about coordination.

* Sentence deleted as recommended

Figure 2 is very blurry 

* This figure has now been revised to improve its clarity.

---

## [Editor Report · Decision Letter 3]

9 Jun 2020

Evaluating the implementation of an Early Supported Discharge (ESD) program for stroke survivors: A mixed methods longitudinal case study

PONE-D-19-26093R3

Dear Dr. Hitch,

We’re pleased to inform you that your manuscript has been judged scientifically suitable for publication and will be formally accepted for publication once it meets all outstanding technical requirements.

Kind regards,

Maw Pin Tan, M.D.

Academic Editor

PLOS ONE
---

## [Editor Report · Acceptance letter]

12 Jun 2020

PONE-D-19-26093R3 

Evaluating the implementation of an Early Supported Discharge (ESD) program for stroke survivors: A mixed methods longitudinal case study 

Dear Dr. Hitch:

I'm pleased to inform you that your manuscript has been deemed suitable for publication in PLOS ONE. Congratulations! Your manuscript is now with our production department. 

Kind regards, 

on behalf of

Dr. Maw Pin Tan 

Academic Editor

PLOS ONE